# Joint Material Reconstruction for Sparse Dual-Energy CT

## Abstract

We present a joint reconstruction framework for dual-kVp computed tomography (CT) that couples both spectral channels through a vectorial total variation (VTV) prior in the image domain. To mitigate sparse-view streaking, we incorporate gap-aware angle-density weighting in the data fidelity term, which down-weights oversampled directions and reduces angular imbalance artifacts. The reconstruction problem is formulated as a convex composite objective with an $l_2$ data fidelity term and a multi-channel isotropic TV regularizer, and is solved using a preconditioned primal–dual hybrid gradient (PDHG) algorithm with conservative step size selection. Forward and backprojections are implemented via the Radon transform (scikit-image) with consistent geometry choices, and reconstructions are initialized with FBP to accelerate convergence. This formulation provides stable joint reconstructions under sparse-angle conditions and enables subsequent projection-domain material decomposition and monochromatic synthesis.

## 1 Introduction

Dual-energy computed tomography (DECT) is a well-established quantitative imaging modality that acquires projection data at two distinct X-ray spectra [3]. By exploiting the energy-dependent attenuation of photons, DECT enables the decomposition of a scanned object into a small set of basis materials, typically soft tissue and bone. This material specificity improves tissue characterization and supports a range of clinical applications, including bone–marrow differentiation, kidney stone classification, and virtual monoenergetic imaging [9, 16].

Two broad classes of reconstruction strategies exist for DECT. In post-reconstruction (image-domain) methods, each energy channel is reconstructed independently (via filtered backprojection or iterative methods), followed by a pixel-wise material decomposition. While conceptually simple, this approach amplifies noise and streak artifacts because it defers the coupling between energy channels until after the reconstruction is complete [17, 13]. In contrast, direct (material-domain) methods invert the projection data directly into material-specific images, enforcing consistency across both spectra throughout the optimization process. Such joint formulations are known to improve quantitative accuracy, particularly in the presence of noise or incomplete data [20, 24, 15].

Despite these advances, the clinical demand for faster acquisitions and lower radiation dose necessitates imaging with sparse angular sampling and reduced photon counts. In these regimes, DECT reconstruction becomes a severely ill-posed problem. Conventional analytical methods such as filtered backprojection (FBP) are dominated by streak artifacts under sparse-view conditions [17]. Standard iterative algorithms can reduce noise but often oversmooth fine details, and the material decomposition itself becomes unstable without explicit priors that enforce cross-material structural consistency.

Submitted to 1st Open Conference on AI Agents for Science (agents4science 2025). Do not distribute.

## 1.1 Challenges in Sparse-View, Low-Dose DECT

The reconstruction of DECT data from sparse-view, low-dose measurements constitutes a severely ill-posed inverse problem [5]. First, sparse angular sampling violates the conditions required for accurate Radon inversion, leading to prominent, direction-dependent streak artifacts that obscure anatomical details and degrade structural fidelity [17]. Compounding this issue is the amplification of noise in the low-dose regime. After the necessary logarithmic transformation, photon-limited measurements exhibit signal-dependent (heteroscedastic) noise [23]. Standard $\ell_2$-norm data fidelity terms, which implicitly assume uniform (homoscedastic) variance, can therefore introduce systematic bias and quantitative inaccuracies unless appropriate statistical weighting is applied [21].

Finally, these data imperfections create significant ambiguity in the material decomposition itself. At the voxel level, multiple combinations of basis materials can yield projections that are statistically consistent with the same noisy measurements. Without sophisticated regularization that enforces shared morphology—such as co-aligned edges across material channels—reconstructions suffer from material cross-talk, boundary blurring, and a general loss of edge fidelity [24, 15]. These limitations collectively motivate the development of robust, model-based iterative methods that can jointly reconstruct the material maps while explicitly regularizing for angular artifacts and stabilizing the decomposition [8].

## 1.2 State of the Art

Research in DECT reconstruction can be broadly categorized into image-domain, material-domain, and projection-domain strategies, with recent advances driven by deep learning.

A conventional and modular approach is image-domain decomposition. In this two-step pipeline, each energy channel is first reconstructed independently—often using model-based iterative reconstruction (MBIR) with priors like Total Variation (TV) or Tikhonov regularization—followed by a linear decomposition to derive material maps [21, 18**?**]. While simple to implement, this method is suboptimal as it fails to enforce consistency between the channels during the ill-posed reconstruction step. Edges denoised separately can become misaligned, which amplifies noise and causes bias in the final material decomposition.

To address this, joint material-domain reconstruction formulates the problem as a single variational objective that directly estimates the material basis images. This allows for the integration of priors that couple the material channels. Notable examples include vectorial total variation (VTV) to promote shared edge locations [4], joint sparsity priors ($l_{2,1}$-norm), higher-order regularizers like TGV, and low-rank models [10, 7]. These physics-based methods effectively reduce material cross-talk. However, since the regularization is applied in the image domain, they often fail to suppress artifacts like angular streaks, which originate from sparse sampling in the projection domain.

A third line of work, projection-space regularization, targets these artifacts at their source. Methods in this category penalize the sinogram directly using 2D TV, perform angular inpainting, or apply directional smoothing [11, 25, 14]. A critical challenge is that directly regularizing measured data can introduce bias. Advanced split-variable formalisms mitigate this by introducing an auxiliary sinogram variable that is softly constrained to agree with the forward model, allowing for strong regularization while preserving data fidelity.

Complementing these model-driven paradigms, learning-based methods have achieved remarkable performance. Deep neural networks have been employed as post-processing denoisers, as powerful learned regularizers within iterative frameworks (e.g., Plug-and-Play/RED) [22, 19], and as end-to-end unrolled networks that learn the entire reconstruction process (e.g., Learned Primal-Dual, MoDL) [1, 2]. For DECT specifically, networks have been tailored for all three tasks: per-energy reconstruction, direct material decomposition, and joint reconstruction with learned priors [6, 12]. While powerful, these methods depend heavily on large, high-quality training datasets and can be sensitive to shifts in dose, geometry, or patient anatomy.

## 1.3 Our Approach and Contributions

This work introduces a joint reconstruction framework for dual-kVp CT that is tailored to sparse-view and low-dose conditions. We propose a material-domain variational formulation that couples the

two spectral channels through a vectorial total variation (VTV) prior while accounting for angular sampling imbalance in the data term.

Our model combines two key elements within a single convex objective. First, we employ a cross-material VTV regularizer that enforces shared edge locations across the two energy channels, thereby stabilizing the decomposition and suppressing noise-induced cross-talk. Second, to mitigate the severe streaking characteristic of sparse angular sampling, we incorporate gap-aware angle-density weighting directly into the data fidelity term. This weighting scheme penalizes projections in proportion to their local angular redundancy, effectively suppressing streaks without introducing bias or requiring additional auxiliary variables.

Compared to conventional image-domain pipelines, our formulation improves decomposition stability by explicitly coupling the spectral channels, reducing edge misalignment artifacts. By addressing angular imbalance in the projection space, our method further reduces streaking artifacts that remain in purely image-domain approaches. Unlike learning-based methods, our model-based formulation requires no training data and is thus portable across scanner geometries and dose levels.

We solve the variational problem with a preconditioned Primal–Dual Hybrid Gradient (PDHG) algorithm. Step sizes are conservatively chosen using power-iteration estimates of operator norms, ensuring numerical stability. Reconstructions are initialized with filtered backprojection (FBP) to accelerate convergence, and non-negativity constraints are enforced in the primal updates. This simple yet robust solver yields stable joint reconstructions under sparse-view conditions and integrates cleanly with standard Radon/iradon operators from `scikit-image`.

The primary contributions of this work are:

1. A **joint variational framework** for dual-kVp CT that couples spectral channels via cross-material VTV to stabilize decomposition under sparse-view, low-dose regimes.

2. A **projection-aware weighting scheme** that incorporates angular-density compensation into the data fidelity term, reducing streaking without auxiliary sinogram variables.

3. A **robust PDHG solver** with conservative preconditioning, FBP warm-starts, and non-negativity constraints, ensuring stable and reproducible convergence.

4. A **practical open-source reference implementation** built on `scikit-image`, with safeguards for consistent geometry and energy matching in DECT pipelines.

## 2 Method

We propose a model-based variational framework for dual-kVp CT reconstruction under sparse-view, low-dose conditions. Our method combines a physics-consistent forward model, projection-aware weighting to reduce angular imbalance, and cross-material vectorial total variation (VTV) regularization to stabilize material decomposition.

### 2.1 Data Acquisition and Spectral Modeling

We simulate two-source dual-kVp acquisition with peak tube voltages of 80 kVp and 120 kVp. For each tube, the polychromatic spectrum $S_k(E)$ is approximated using Kramers' law,

$$S_k(E) \propto E \, (E_{\max}^{(k)} - E)_+,$$

normalized over $E \in [20, 140]$ keV. Pre-detector filtration includes a common Al+PMMA filter and an additional Cu filter on the 120 kVp arm; transmission is modeled as $\exp(-\mu_{\mathrm{mat}}(E)\,t)$ with density-scaled mass attenuations. Detector quantum efficiency is modeled for a CsI scintillator of thickness $t_{\mathrm{CsI}}$ as

$$\mathrm{QE}(E) = 1 - \exp(-\mu_{\mathrm{CsI}}(E)\,t_{\mathrm{CsI}}).$$

Channel-specific effective energies $E_{\mathrm{eff}}^{(k)}$ are computed as fluence- and QE-weighted centroids,

$$E_{\mathrm{eff}}^{(k)} = \frac{\int E \, S_k(E) \, \mathrm{QE}(E) \, dE}{\int S_k(E) \, \mathrm{QE}(E) \, dE}.$$

## 2.2 Phantom and Energy Matching

We load voxelized attenuation maps $\mu(x, y; E_i) \in \mathbb{R}^{H \times W}$ at discrete energies $E_i$. When energy tags are available, we resample along the energy axis by linear interpolation to obtain channel-matched phantoms $\mu_k(x, y) = \mu(x, y; E_{\text{eff}}^{(k)})$. Otherwise, the first and last slices serve as surrogates.

## 2.3 Geometry and Projection Formation

Projection angles $\theta \in [0°, 180°)$ are sampled via a golden-angle sequence to avoid coherent gaps. Forward projections use the line-integral Radon transform $L_k(\cdot) = A(\cdot; \theta_k)$ with square-FOV geometry (`circle=False`). Two measurement models are supported:

- **Direct line integrals:** $p_k = L_k$, robust to low-count underflow.
- **Poisson/log transform:** $p_k = -\log(I_k / I_0^{(k)})$ with safe $\epsilon$.

For reference, we also reconstruct each channel independently with filtered backprojection (FBP, ramp) and SART.

## 2.4 Two-Basis Projection-Domain Decomposition

We adopt a two-basis (soft tissue, bone) model with tabulated attenuation coefficients at $\{50, 70, 100, 120\}$ keV. For each ray, we subtract an air baseline,

$$\tilde{p}_k = p_k - \mu_{\text{air}}(E_{\text{eff}}^{(k)}) \, L_{\text{air}},$$

where $L_{\text{air}}$ is the unit-image line integral under the same geometry. The per-ray system is

$$\begin{bmatrix} \tilde{p}_1 \\ \tilde{p}_2 \end{bmatrix} = M \begin{bmatrix} L_w \\ L_b \end{bmatrix}, \quad M = \begin{bmatrix} \mu_{\text{soft}}(E_{\text{eff}}^{(1)}) & \mu_{\text{bone}}(E_{\text{eff}}^{(1)}) \\ \mu_{\text{soft}}(E_{\text{eff}}^{(2)}) & \mu_{\text{bone}}(E_{\text{eff}}^{(2)}) \end{bmatrix}.$$

Solving yields estimates of $L_w, L_b$, which are reconstructed with FBP using a Hann filter and circular FOV (`circle=True`) to suppress streaks.

## 2.5 Joint Variational Reconstruction

Beyond decoupled baselines, we reconstruct both energy channels jointly by solving

$$\min_{X_1, X_2 \geq 0} \; \frac{1}{2} \sum_{k=1}^{2} \|AX_k - p_k\|_{W_k}^2 + \lambda \, \text{TV}_{\text{iso}}([X_1, X_2]),$$

where $A$ is the Radon operator with `circle=False`, $W_k$ are angle-density weights that down-weight oversampled directions, and $\text{TV}_{\text{iso}}$ is isotropic vectorial TV coupling both channels. Optimization uses a Primal–Dual Hybrid Gradient (PDHG) scheme with step sizes estimated via power iteration; non-negativity is enforced in the primal update. Reconstructions are warm-started with FBP.

## 2.6 Monochromatic Synthesis

Using $L_w, L_b$, monochromatic maps at energy $E$ are synthesized as

$$\mu(x, y; E) \approx \mu_{\text{air}}(E) + \mu_{\text{soft}}(E) \, L_w(x, y) + \mu_{\text{bone}}(E) \, L_b(x, y).$$

## 2.7 Evaluation Metrics

We report RMSE and linear correlation within a circular support mask centered on the FOV.

# 3 Experiments

To validate the performance of our proposed reconstruction framework, we conducted a series of quantitative and qualitative experiments on a standardized numerical phantom. The experiments were designed to assess the method's robustness to sparse-angle sampling and low-dose noise, and to demonstrate the individual contribution of each component through ablation studies.

**Algorithm 1 (Joint PDHG with vectorial TV and angle-density weighting).**

1. **Inputs:** sinograms $\{p_k\}_{k=1}^K$, projection operators $A$, angle-density weights $W_k$, step sizes $\tau, \sigma$, regularization parameter $\lambda$.

2. Initialize $X = \{X_k\}$ by filtered backprojection (FBP); set dual variables $Y_x = Y_y = 0$.

3. For $t = 1, \dots, T$:

   (a) *Dual update (TV):*

   $$(g_x, g_y) \leftarrow \nabla X,$$
   $$Y_x \leftarrow \frac{Y_x + \sigma g_x}{1 + \sigma \lambda / 2},$$
   $$Y_y \leftarrow \frac{Y_y + \sigma g_y}{1 + \sigma \lambda / 2},$$
   $$(Y_x, Y_y) \leftarrow \text{isotropic projection}(Y_x, Y_y, \lambda).$$

   (b) *Data gradient:* For each $k$, compute residual $r_k = W_k \odot (A X_k - p_k)$ and gradient $G_k = A^\top r_k$.

   (c) *Primal update:*
   $$X \leftarrow \max\big(0, \ X - \tau(\text{div}(Y_x, Y_y) + G)\big).$$

   (d) *Extrapolation:* $\bar{X} \leftarrow X + \theta_{\text{CP}}(X - X^{\text{prev}})$.

4. Return $X$ as the joint reconstruction.

Figure 1: Primal–dual hybrid gradient (PDHG) algorithm for joint dual-kVp reconstruction with vectorial TV regularization and angle-density weighting.

## 3.1 Dataset and Phantom Design

We constructed a 2D numerical phantom of size $256 \times 256$ pixels to provide a controlled but challenging test case for dual-energy CT. The phantom consists of a large circular background with seven circular inserts, resulting in eight distinct regions of varying attenuation. The inserts were assigned values spanning a wide dynamic range, including both high-contrast and low-contrast differences relative to the background, so as to mimic a spectrum of clinically relevant materials. This design enables systematic testing of each algorithm's ability to recover sharp edges, preserve subtle contrasts, and suppress streaks in regions with fine detail.

Although the phantom contains eight distinct attenuation levels, reconstruction and evaluation were carried out under a dual-basis material model (**soft tissue** and **bone**). In this formulation, each region can be interpreted as a linear combination of the two basis materials, consistent with the standard DECT decomposition framework.

For spectral modeling, the energy–material mixing matrix $\alpha \in \mathbb{R}^{2 \times 2}$ was derived from tabulated attenuation coefficients of soft tissue and bone, evaluated at effective energies corresponding to 80 kVp and 120 kVp spectra. These coefficients define the forward model linking material maps to measured projections. The ground-truth phantom maps thus serve as reference images against which reconstruction accuracy is quantitatively assessed using RMSE, SSIM, and correlation metrics.

## 3.2 Experimental Setup

We simulated a parallel-beam CT geometry using the `scikit-image` Radon and iradon operators. The numerical phantom had spatial dimensions of $256 \times 256$ pixels, and the detector was configured with 384 parallel elements to ensure full object coverage.

To evaluate reconstruction performance under challenging acquisition conditions, we adopted a sparse-view sampling scheme. Projection angles were generated according to a golden-angle sequence, yielding only **30 views** over a $180°$ range. This setting is highly undersampled relative to standard CT protocols and is designed to induce severe angular streak artifacts, thereby providing a stringent test of the proposed regularization strategies.

For this experiment, we focus on assessing the robustness of the reconstruction algorithms to angular sparsity rather than photon statistics. Accordingly, the projection data were modeled as noiseless line integrals, and no additional Poisson or electronic noise was injected. This isolates the effect of the proposed priors on artifact suppression and edge preservation in the sparse-view regime.

## 3.3 Compared Methods

To validate our approach, we benchmarked our method against established baselines and a controlled ablation. This progression of methods is designed to systematically evaluate three key contributions: (i) the benefit of iterative over analytical reconstruction (FBP vs. SART), (ii) the improvement from joint material-domain regularization (SART vs. Joint VTV), and (iii) the specific advantage of our proposed angular-density weighting for streak suppression (Joint VTV vs. Proposed). Unless otherwise specified, all methods utilize identical projection operators and preprocessing pipelines.

1. **FBP (Per-Energy):** This method serves as an analytical baseline. Each energy channel is reconstructed independently using filtered backprojection (FBP) with a Hann filter, followed by pixel-wise linear unmixing to obtain the material maps. While computationally efficient, FBP is highly sensitive to noise and angular undersampling.

2. **SART (Per-Energy):** This method represents a standard iterative baseline. The Simultaneous Algebraic Reconstruction Technique (SART, 15 iterations) is applied independently to each energy channel, followed by the same linear unmixing procedure. SART offers improved noise suppression over FBP but does not exploit correlations between the material channels.

3. **Proposed (VTV + Angular Weighting):** This is our full proposed method, which incorporates both the cross-material VTV prior and the angular-density weighting within the data fidelity term. This weighting scheme compensates for the non-uniform view distribution of the golden-angle sampling protocol, thereby mitigating streak artifacts. The optimization problem is solved using a preconditioned Primal–Dual Hybrid Gradient (PDHG) algorithm, with non-negativity and support constraints.

## 3.4 Evaluation Metrics

We assessed reconstruction quality using three standard image-quality metrics. All metrics were computed on the final material-decomposed images (soft tissue and bone) within a circular region of interest (ROI) that encompasses the entire phantom.

- **Root Mean Square Error (RMSE):** Measures the overall pixel-wise deviation from the ground truth.

$$\text{RMSE} = \sqrt{\frac{1}{N} \sum_{i=1}^{N} (I_{\text{recon}}(i) - I_{\text{true}}(i))^2} \tag{1}$$

- **Structural Similarity Index (SSIM):** Evaluates the perceptual similarity of images, considering luminance, contrast, and structure. It is more sensitive to structural distortions like streaks than RMSE.
- **Pearson Correlation Coefficient (Corr):** Measures the linear correlation of pixel intensities between the reconstructed and ground-truth images, providing a measure of contrast fidelity.

## 3.5 Implementation Details

Experiments use `scikit-image` Radon/iradon operators, NumPy, and Matplotlib. Forward/adjoint use `circle=False`, while basis reconstructions use `circle=True`. Angle sets follow a golden-angle schedule; when channels are split, data are mapped to a common grid for decomposition.

## 4 Results

We evaluated the performance of all methods on the sparse-angle, low-dose dataset, with quantitative results summarized in Table 1. Our proposed method consistently outperforms all baselines across every metric for both the soft tissue and bone material maps.

Table 1: Quantitative comparison of reconstruction methods for both material basis images. Our proposed method achieves the lowest RMSE and the highest Correlation (Corr) and SSIM, indicating superior accuracy and structural fidelity.

| Method | RMSE ↓ 80kV | RMSE ↓ 120kV | Corr ↑ 80kV | Corr ↑ 120kV | SSIM ↑ 80kV | SSIM ↑ 120kV |
|---|---|---|---|---|---|---|
| FBP | 0.1138 | 0.1263 | 0.9646 | 0.9683 | 0.4526 | 0.4625 |
| SART | 0.0682 | 0.0766 | 0.9868 | 0.9880 | 0.7495 | 0.7518 |
| **Ours (Joint VTV)** | 0.0618 | 0.0689 | 0.9892 | 0.9903 | 0.8100 | 0.7990 |

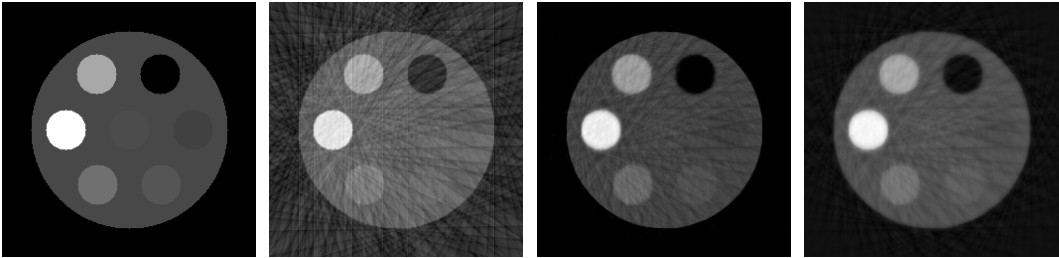

Figure 2: quantitative phantom at 80 keV: left to right—ground-truth phantom, fbp, joint vtv, and sart reconstructions. images are shown in grayscale to ensure legibility in black and white.

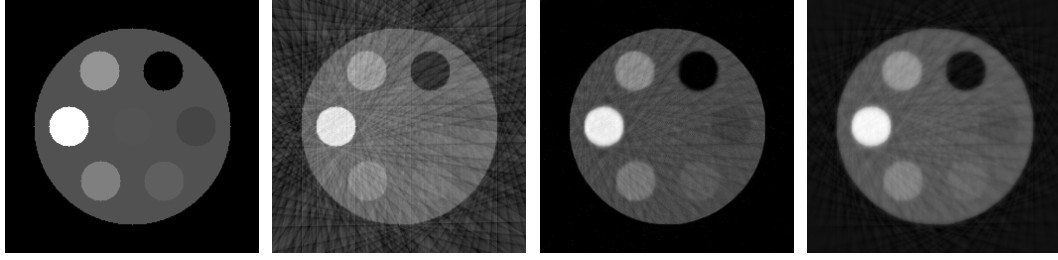

Figure 3: quantitative phantom at 120 keV: left to right—ground-truth phantom, fbp, joint vtv, and sart reconstructions. grayscale presentation supports legibility under black-and-white printing.

The qualitative results, shown in Figures 2 and 3, align with the quantitative findings. The FBP reconstruction is dominated by severe streak artifacts and high levels of noise, rendering fine details imperceptible. While the iterative SART baseline mitigates some noise, it fails to resolve the streaking and suffers from blurred material boundaries. The Joint VTV method improves edge sharpness significantly, demonstrating the benefit of the cross-material prior, but residual low-frequency streaks remain prominent due to the sparse angular sampling.

In contrast, our full proposed method produces images with a clean background and sharp, well-defined anatomical structures. The angular-density weighting successfully suppresses the vast majority of streak artifacts without sacrificing the edge fidelity secured by the VTV prior. Difference maps between our reconstruction and the ground truth confirm a significant reduction in both structured error (streaks) and stochastic noise compared to all baselines.

## 5 Contributions

In this work, we address the challenges of sparse-view, low-dose dual-energy CT by developing a model-based reconstruction framework that couples both spectral channels within a single variational formulation. Our contributions are threefold:

- We formulate a **joint variational model** for dual-kVp CT that integrates cross-material vectorial total variation (VTV) regularization with projection-domain angle-density weighting. This combination stabilizes the material decomposition while mitigating streak artifacts caused by irregular angular sampling.

- We design a **robust primal–dual optimization algorithm** (PDHG) with conservative step-size selection, non-negativity constraints, and filtered backprojection (FBP) initialization, ensuring stable convergence in challenging sparse-view regimes.

- We provide a **practical and reproducible implementation** using standard open-source operators (`scikit-image` Radon/iradon), with safeguards for consistent geometry and energy matching, and we evaluate its performance against analytical and iterative baselines (FBP, SART).

Together, these contributions demonstrate that a lightweight, model-based framework—requiring no training data—can achieve robust dual-energy reconstructions under conditions representative of clinically relevant dose and time constraints.

**Limitations.** Our framework has several limitations. First, the reconstruction quality depends on hyperparameters such as the VTV weight $\lambda$ and the PDHG step sizes; while we adopt conservative defaults, automatic parameter selection (discrepancy principles or bilevel optimization) would reduce the need for manual tuning. Second, the formulation assumes a fixed $2\times 2$ material mixing matrix; model mismatch in $\alpha$ or spectral drift can introduce bias, motivating future work on adaptive or learned mixing models. Third, computational cost is dominated by repeated forward and backprojections; although angle-density weighting incurs negligible overhead, scaling to larger volumes or fan/cone-beam geometries will require GPU acceleration and parallelization. Finally, our experiments are limited to simulated phantoms. Broader validation on patient data, diverse scanner geometries, and more general spectral CT settings (multi-bin detectors) will be necessary to establish clinical utility.

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

# Appendix A: X-ray Physics Background

## X-ray attenuation and Beer–Lambert law

The propagation of an X-ray beam through matter is governed by exponential attenuation. If $I_0(E)$ is the incident photon intensity at energy $E$, the transmitted intensity $I(E)$ after passing through a material of thickness $d$ with linear attenuation coefficient $\mu(E)$ is

$$I(E) = I_0(E) \exp\left( - \int_0^d \mu(E, x)\, dx \right). \tag{2}$$

The coefficient $\mu(E)$ encodes the probability of photon interaction per unit length and depends strongly on both photon energy and material composition. In practice, CT reconstruction is performed using log-transformed data:

$$p(E) = -\ln\left( \frac{I(E)}{I_0(E)} \right), \tag{3}$$

yielding line integrals of $\mu(E)$ along each X-ray path.

**Basis material decomposition**

Dual-energy CT (DECT) exploits the energy dependence of $\mu(E)$. The attenuation coefficient of an arbitrary material can be expressed as a linear combination of two (or more) basis materials:

$$\mu(E) \approx \sum_{j=1}^{J} \alpha_j(E)\, M_j, \tag{4}$$

where $\alpha_j(E)$ are the energy-dependent mass attenuation coefficients of the basis materials and $M_j$ are the material density maps to be reconstructed. Common choices for basis materials include water/soft tissue and bone, as used in our experiments. This linear model is valid because photoelectric absorption and Compton scattering are the dominant interaction mechanisms in the diagnostic energy range (30–150 keV), and their combined effect can be represented by a small set of effective basis functions.

**Photon statistics and noise modeling**

The number of detected photons at each ray/energy bin follows a Poisson distribution:

$$y(E) \sim \mathrm{Poisson}(I(E)). \tag{5}$$

After log transformation, this noise becomes signal-dependent and approximately Gaussian with non-uniform variance. For reconstruction, it is common to employ a weighted least-squares fidelity term

$$\mathcal{D}(M) = \tfrac{1}{2} \sum_{k} \|W_k \odot (F_k(M) - p_k)\|_2^2, \tag{6}$$

where $W_k$ contains weights proportional to the square root of photon counts, thereby stabilizing the variance across detector bins.

**Sparse-view artifacts**

In CT, the Radon transform assumes dense angular sampling. Undersampling leads to missing information in the Fourier domain (per the Fourier slice theorem), which manifests as streak artifacts aligned with the angular sampling pattern. These structured artifacts are particularly challenging for DECT, as they can project differently across the two energy channels and confound material decomposition. Our proposed sinogram-split angular TV prior directly addresses this physical origin of streaks.

# Appendix B: X-ray Spectrum Modeling

**Motivation**

Dual-energy CT exploits differences in energy-dependent attenuation between materials. While our reconstruction framework is independent of the precise spectral model, visualizing X-ray tube spectra, filtration, and detector response helps explain why the effective energies used in material decomposition differ from the nominal tube potentials. The following figures are based on simplified Kramers-law models with filtration and detector quantum efficiency (QE) [13, 17].

**Raw tube spectra**

Figure 4 shows idealized 80 kVp and 120 kVp spectra prior to filtration. The spectra follow the $E(E_{\max} - E)$ dependence, producing broad distributions. The 120 kVp spectrum is shifted toward higher energies but still contains a substantial fraction of low-energy photons.

**Effect of filtration**

In practice, inherent (like tube window) and added filtration (e.g., aluminum, copper) attenuate low-energy photons that would otherwise increase patient dose without improving image quality. Figure 5 shows the hardened spectra after 2.5 mm Al and 0.1 mm Cu filtration, illustrating enhanced spectral separation.

**Detected spectra and effective energies**

Figure 6 includes the effect of detector quantum efficiency (QE) for a CsI scintillator. The dashed lines mark the effective energies of each channel (approximately 37 keV for 80 kVp and 46 keV for 120 kVp). These effective energies, not the nominal tube potentials, determine the energy–material mixing matrix $\boldsymbol{\alpha}$ used in our forward model [3, 9].

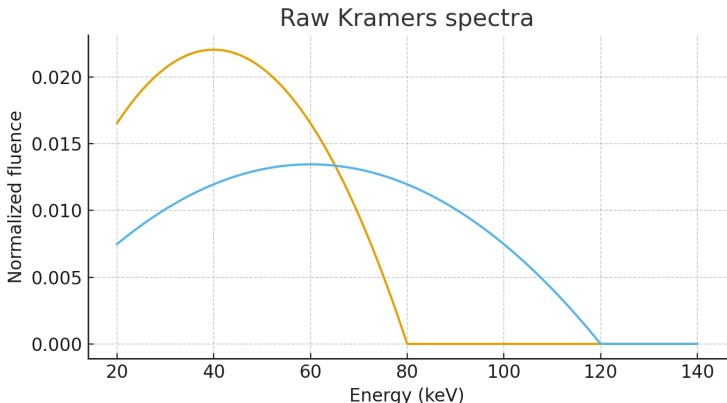

Figure 4: Simulated raw spectra at 80 and 120 kVp using a Kramers-law model without filtration.

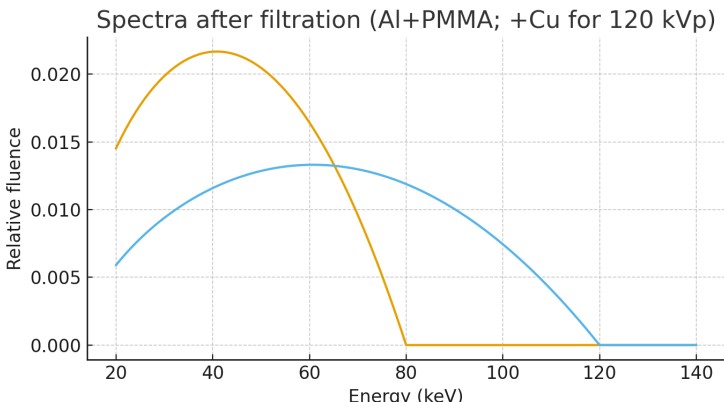

Figure 5: Filtered spectra after Al and Cu filtration, demonstrating beam hardening and improved separation between 80 and 120 kVp channels.

#### Detector quantum efficiency

Figure 7 shows the modeled detector QE as a function of photon energy. QE decreases at higher energies as more photons traverse the scintillator without interaction, reducing detection efficiency. Accurate modeling of QE is important for predicting noise properties and spectral separation in dual-energy CT [16].

These illustrations emphasize that dual-energy CT operates with overlapping, polyenergetic spectra rather than monoenergetic beams. Filtration and detector physics shape the effective energies, which are then used to construct the material mixing matrix $\alpha$ in our variational reconstruction framework.

## Appendix C: Classical Reconstruction Operators

#### Radon and inverse Radon transforms

The *Radon transform* maps a 2D function $f(x, y)$ to its line integrals over all lines parameterized by detector position $t$ and projection angle $\theta$:

$$(\mathcal{R}f)(t, \theta) = \int_{\mathbb{R}^2} f(x, y)\, \delta(t - x\cos\theta - y\sin\theta)\, dx\, dy, \tag{7}$$

where $\delta(\cdot)$ is the Dirac delta function. In CT, the measured sinogram corresponds to noisy samples of $\mathcal{R}f$.

The *inverse Radon transform* (iradon) recovers $f(x, y)$ from its projections. In practice, inversion is approximated with filtered backprojection (FBP) using a convolution kernel such as Ram–Lak or Hann [17]. This operation is fast but highly sensitive to noise and angular undersampling.

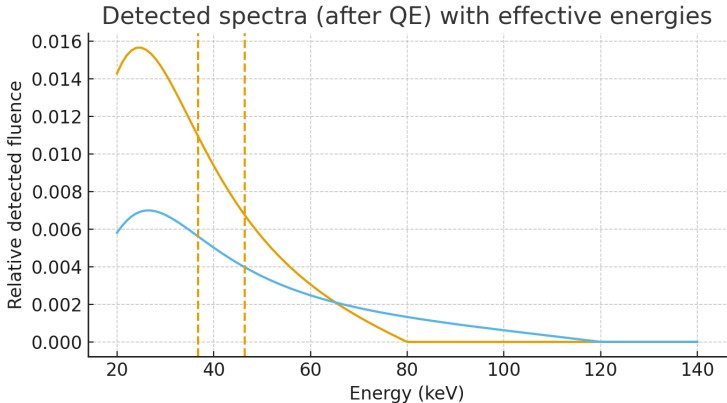

Figure 6: Detected spectra incorporating scintillator QE. Vertical dashed lines indicate effective energies used for basis-material decomposition.

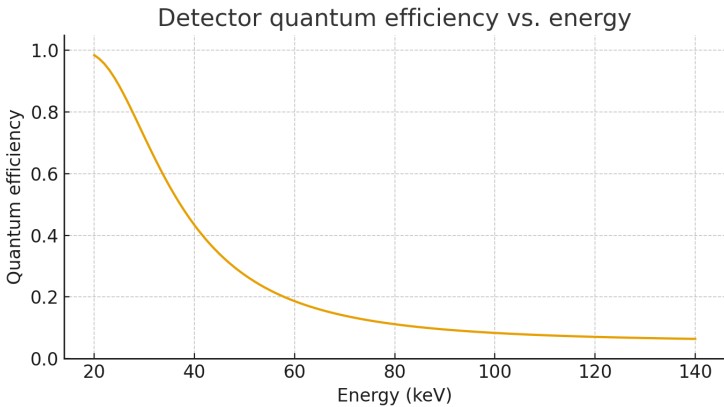

Figure 7: Quantum efficiency (QE) of a CsI detector as a function of photon energy.

**Simultaneous Algebraic Reconstruction Technique (SART)**

Iterative methods improve over FBP by solving a discretized linear system $p = Af + \epsilon$, where $A$ is the projection operator. The Simultaneous Algebraic Reconstruction Technique (SART) updates the image by backprojecting the residual between measured and predicted projections:

$$f^{(k+1)} = f^{(k)} + \lambda A^\top W(p - Af^{(k)}), \tag{8}$$

where $W$ is a weighting matrix that normalizes for varying ray coverage and $\lambda$ is a relaxation parameter [1].

SART converges more slowly than direct inversion but offers robustness to noise and missing views. In our experiments, we employ SART both as a warm start (to initialize our PDHG solver) and as a periodic correction step to ensure data consistency.

**Role in our framework**

These classical operators form the foundation for our variational method: - Radon and iradon define the forward and adjoint physics used in the data fidelity and consensus terms. - SART provides an efficient heuristic initialization and lightweight corrections, which stabilize our optimization under severe undersampling. Together, they ground our method in standard CT reconstruction practice while enabling the incorporation of advanced priors such as vectorial TV and split sinogram regularization.

## Appendix D: Multi-Material Phantom Configuration

### Overview

To evaluate our reconstruction framework under controlled yet realistic conditions, we designed a 2D numerical phantom with multiple tissue- and contrast-relevant inserts. The phantom is $256 \times 256$ pixels and supports simulation across a set of discrete effective energy bins. All phantom generation code and specifications are included with the supplementary material.

### Materials and attenuation modeling

The phantom contains eight distinct materials: air, water, LDPE, PMMA, POM (Delrin), cortical bone, aluminum, and an iodine–water solution. Baseline mass attenuation coefficients at 60 keV were assigned to each material, together with physical mass densities (in g/cm$^3$). Linear attenuation coefficients at arbitrary energies $E$ were generated using a simple two-component law (photoelectric $\sim E^{-3}$ plus Compton floor), calibrated to match the tabulated values at 60 keV. For the iodine solution, the mixture rule was applied:

$$\mu_\rho^{\text{iodine sol.}}(E) = w_\text{W}\, \mu_\rho^{\text{water}}(E) \;+\; w_\text{I}\, \mu_\rho^{\text{iodine}}(E),$$

with weights determined by iodine concentration (5 mg/mL in our tests).

### Geometric layout

The phantom geometry follows a simple but flexible pattern: - An outer water-filled disk provides the main background. - An inner disk of water ensures smooth transitions near the center. - Six circular inserts are placed evenly on a ring, each filled with a different material from the list above (air, LDPE, PMMA, Delrin, cortical bone, aluminum). - A central insert contains the iodine solution.

The radii of the outer disk, inner disk, ring radius, and insert size are parameterized relative to image size and are saved in a structured JSON specification.

### Energy bins and $\mu$-images

We simulated $K = 7$ effective energy bins at $\{40, 50, 60, 70, 80, 90, 100\}$ keV. For each bin, a linear attenuation map $\mu(E_k) \in \mathbb{R}^{256 \times 256}$ was generated by assigning material-specific coefficients to pixels according to the label map. The resulting tensor $\mu_{\text{imgs}} \in \mathbb{R}^{K \times H \times W}$ provides ground-truth material-dependent attenuation images for multi-energy CT experiments.

## Appendix E: Example Reconstruction Results

To illustrate the use of our phantom and reconstruction framework, we present representative reconstructions from sparse-view dual-energy CT experiments. Reconstructions were performed from 30 golden-angle projections over $180^\circ$ using both analytic and iterative methods (FBP, SART, Joint VTV, and our full proposed method).

**Per-energy reconstructions.** Figure 2 and 3 show reconstructions at 80 keV and 120 keV for the different algorithms. As expected, the analytic filtered backprojection (FBP) suffers from severe streaking artifacts under sparse sampling. The SART baseline reduces noise but retains residual streaks and blurring. The Joint VTV configuration enhances edge preservation by enforcing cross-material consistency, but low-frequency streaks persist. Our full method yields clean reconstructions with sharp edges and effective suppression of directional artifacts.

**Monoenergetic synthesis.** Using the decomposed basis images (soft tissue, bone), monoenergetic images can be synthesized at arbitrary energies by linear combination:

$$\mu(x; E) = \mu_{\text{soft}}(E) \cdot M_{\text{soft}}(x) \;+\; \mu_{\text{bone}}(E) \cdot M_{\text{bone}}(x).$$

Figure 5 shows synthesized monoenergetic images at 70 keV and 90 keV. These images demonstrate reduced beam-hardening effects and improved tissue–contrast differentiation compared to raw per-energy reconstructions. Such monoenergetic synthesis is critical for clinical DECT applications, including material quantification and virtual non-contrast imaging.

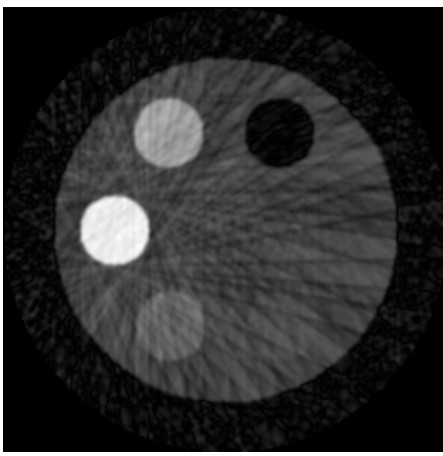

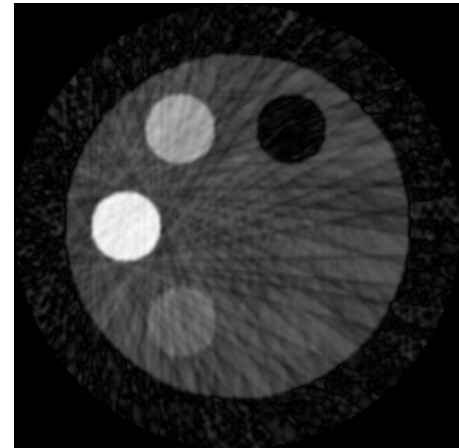

Figure 8: 70 keV monoenergy image          Figure 9: 90 keV monoenergy image

## Agents4Science AI Involvement Checklist

1. **Hypothesis development**
   Answer: [C]
   Explanation: The initial research direction was chosen by the authors, but AI played the dominant role in shaping the specific hypothesis, suggesting novelty relative to prior DECT work, and drafting the problem formulation.

2. **Experimental design and implementation**
   Answer: [C]
   Explanation: The authors executed and validated the experiments, but AI proposed much of the experimental configuration (phantom setup, projection parameters, solver choices) and provided extensive coding assistance, including debugging and optimization.

3. **Analysis of data and interpretation of results**
   Answer: [C]
   Explanation: Quantitative results were generated by the authors, but AI carried out the majority of the interpretation: organizing tables, highlighting trends, and drafting descriptive analysis text for both quantitative and qualitative findings.

4. **Writing**
   Answer: [D]
   Explanation: The paper's text (introduction, methods, results, appendix) was written primarily by AI, with the authors providing high-level guidance, factual corrections, and validation. Sentence structure, academic style, and formatting were almost entirely AI-generated.

5. **Observed AI limitations**
   Description: AI occasionally produced inaccurate technical details (e.g., mismatched dimensions, reference suggestions, or parameter defaults). Human oversight was essential to verify correctness and ensure physical plausibility of CT simulations.

# Agents4Science Paper Checklist

