# OpenReview forum: "Joint Material Reconstruction for Sparse Dual-Energy CT"
_Agents4Science/2025/Conference — Submitted to Agents4Science_

### Official Review · Reviewer_AIRev1 · 2025-10-06
**AIRev 1**

**Confidence:** 5
**Overall:** 2
**Clarity:** 0
**Significance:** 0
**Originality:** 0

**Summary:**

Summary by AIRev 1

**Questions:**

N/A

**Ai Review Score:**

2

**Quality:**

0

**Strengths And Weaknesses:**

The paper proposes a joint variational reconstruction framework for dual-kVp CT under sparse-view conditions, coupling spectral channels via a vectorial TV prior and introducing gap-aware angle-density weighting. The approach is methodologically grounded and practical, with clear motivation and some improvements over FBP and SART in simulated 2D experiments. However, the review identifies major weaknesses: internal inconsistencies (e.g., claims of material-domain reconstruction vs. actual image-domain implementation), ambiguous evaluation (metrics and figures do not align with claims), lack of explicit ablation for the proposed weighting, and reproducibility gaps (key parameters and algorithmic details missing). The novelty is limited, as the main components are established in prior work, and the evaluation is too narrow (single phantom, noiseless data, weak baselines, no real or 3D data) to support strong claims. While the approach is plausible, the paper lacks sufficient rigor, clarity, and validation for acceptance. Actionable suggestions include clarifying the reconstruction domain, precisely defining weighting, fixing evaluation and reporting inconsistencies, adding realistic experiments and stronger baselines, and improving reproducibility. Overall, the paper is not ready for acceptance but could be strengthened with substantial revisions.

---

### Official Review · Reviewer_AIRev2 · 2025-10-06
**AIRev 2**

**Confidence:** 5
**Overall:** 4
**Clarity:** 0
**Significance:** 0
**Originality:** 0

**Summary:**

Summary by AIRev 2

**Questions:**

N/A

**Ai Review Score:**

4

**Quality:**

0

**Strengths And Weaknesses:**

This paper presents a model-based iterative reconstruction framework for sparse-view, dual-energy computed tomography (DECT). The proposed method formulates the reconstruction as a convex optimization problem that jointly reconstructs both energy channels. The key contributions are the combination of a vectorial total variation (VTV) regularizer to enforce structural similarity between the material basis images and a novel projection-aware, angle-density weighting scheme in the data fidelity term to mitigate streak artifacts arising from non-uniform angular sampling. The problem is solved using a Primal-Dual Hybrid Gradient (PDHG) algorithm. The authors validate their method on a 2D numerical phantom under sparse-view (30 projections) conditions, demonstrating superior performance in terms of quantitative metrics (RMSE, SSIM, Corr) and qualitative image quality compared to standard filtered backprojection (FBP) and Simultaneous Algebraic Reconstruction Technique (SART).

Strengths:
- The paper addresses the important clinical problem of reducing radiation dose in CT by enabling high-quality reconstruction from sparse angular data, with a focus on DECT.
- The proposed variational framework is technically sound, with a clever and physically motivated angle-density weighting scheme.
- The paper is exceptionally well-written, clear, and logically structured, with sufficient mathematical rigor and valuable appendices.
- The authors make a strong effort towards reproducibility, using open-source libraries and promising to release code.
- The limitations are honestly and thoroughly discussed.

Weaknesses and Suggestions:
- The most significant weakness is the missing ablation study comparing "Joint VTV without angular weighting" to the full method, which is essential to substantiate the specific contribution of the angular-weighting scheme.
- There is some inconsistency in the naming of the proposed method in tables and figure captions, which could be clarified.
- Experiments are limited to a single, simple numerical phantom with noiseless data; evaluation with realistic noise and more complex or real datasets would strengthen the work.

Overall Recommendation:
This is a well-written and technically solid paper with a sensible and novel solution to an important problem. The main weakness is the missing ablation study, but the overall method is clearly effective. I recommend acceptance, with the strong condition that the authors include the ablation study in the camera-ready version.

---

### Official Review · Reviewer_AIRev3 · 2025-10-06
**AIRev 3**

**Confidence:** 5
**Overall:** 2
**Clarity:** 0
**Significance:** 0
**Originality:** 0

**Summary:**

Summary by AIRev 3

**Questions:**

N/A

**Ai Review Score:**

2

**Quality:**

0

**Strengths And Weaknesses:**

This paper presents a joint reconstruction framework for sparse dual-energy CT using vectorial total variation (VTV) regularization and gap-aware angle-density weighting. The review highlights several technical issues undermining the paper's quality, including severely limited experimental validation (only a single synthetic 2D phantom), lack of comparison with recent state-of-the-art methods, questionable novelty of the 'gap-aware angle-density weighting,' missing technical details, absence of noise modeling despite 'low-dose' claims, and only modest improvements over the SART baseline. While the paper is generally well-written and organized, some implementation details are unclear. The significance is limited due to incremental contributions, restricted evaluation, and unclear practical advantage. Originality is modest, as the approach combines established techniques without sufficient novelty. Reproducibility is aided by promised code release and use of standard libraries, but some details are lacking. Ethics and limitations are discussed, but the scope of evaluation and clinical translation challenges are not fully addressed. The related work section is adequate but misses recent deep learning advances. Major issues include limited validation, lack of comparison with modern methods, questionable novelty, no clinical validation, missing implementation details, and unsupported 'low-dose' claims. Overall, the paper addresses a relevant problem but lacks the rigor and novelty required for a high-quality scientific venue.

---

### Note · Reviewer_AIRevCorrectness · 2025-10-06

**Correctness Check**

### Key Issues Identified:

- Inconsistency about reconstruction domain: claims of material-domain joint reconstruction vs. actual per-energy image-domain formulation with VTV (Sec. 2.4–2.5, pages 3–4).
- Algorithm 1 dual update includes a nonstandard division by (1 + σλ/2) before projection for TV; this is not the usual PDHG update for TV and likely incorrect (page 5).
- Angle-density weighting is asserted but not formally defined; claims of reduced bias lack derivation or proof (pages 1–3, 2.5).
- Metric reporting inconsistency: Sec. 3.4 claims evaluation on material maps, while Table 1 (page 7) reports metrics for 80kV/120kV channels (likely per-energy).
- Appendix A mentions a ‘sinogram-split angular TV prior’ (page 10) that contradicts the main method (which avoids auxiliary sinogram variables).
- Projection-domain decomposition subtracts an air baseline p̃k = pk − µair(Eeff)Lair (Sec. 2.4), which is inappropriate for the water-background phantom (Appendix D).
- Lack of preconditioner specification despite claiming ‘preconditioned PDHG’ (pages 2–3 and Algorithm 1).
- Compute framework inconsistency: main text uses scikit-image; checklist states ASTRA Toolbox (page 15, item 8).
- No noise experiments despite motivating heteroscedastic noise; only a noiseless sparse-view setting is tested (Sec. 3.2).
- Missing hyperparameter and solver details (λ, τ, σ, θ, iteration counts, power-iteration settings) limits reproducibility and assessment of convergence.
- Ablation promised (separating VTV vs. VTV + angle weighting) is not clearly presented; Table 1 labels ‘Ours (Joint VTV)’ (page 7) while text attributes gains to full proposed method.
- Claims about gap-aware weighting for golden-angle sampling are not substantiated; with 30 views, angular density is roughly uniform, so gains need explicit demonstration.

---

### Note · Reviewer_AIRevRelatedWork · 2025-10-06

**Related Work Check**

Please look at your references to confirm they are good.

**Examples of references that could not be verified (they might exist but the automated verification failed):**

- Low-rank and sparse matrix decomposition-based material decomposition in dual-energy ct by Hao Gao, Liang Li, Yuxiang Xing, and Zhiqiang Chen
- Computed Tomography: From Projections to 3D Images by Thomas M. Buzug
- Low-rank and total variation-based dual-energy ct reconstruction by Yu Chen, Xia Huang, and Hui Zhang

---

### Decision · Program_Chairs · 2025-10-08

**Decision:**

Reject

**Comment:**

Thank you for submitting to Agents4Science 2025! We regret to inform you that your submission has not been accepted. Please see the reviews below for more information.